# Health Professional Support for Friends and Family Members of Older People Discharged from Hospital After a Fracture: A Survey Study

**DOI:** 10.3390/geriatrics10020036

**Published:** 2025-03-07

**Authors:** Toby O. Smith, Susanne Arnold, Mark Baxter

**Affiliations:** 1Warwick Medical School, University of Warwick, Coventry CV4 7AJ, UK; susanne.arnold@warwick.ac.uk; 2Medicine for Older People, University Hospitals Southampton, Southampton SO16 6YD, UK; mark.baxter@uhs.nhs.uk

**Keywords:** carer, hospital discharge, recovery, rehabilitation, spouse, fragility fracture, trauma

## Abstract

**Background/Objectives:** Friends and family members of people who are discharged from hospital after a fracture often take on caring roles, since these patients have reduced independence during recovery. Previous literature suggests that these individuals are rarely supported in their adoption of these roles. No studies have previously explored the use of carer training interventions to support friends/family members by health professionals in this setting. This survey study aimed to address this. **Methods:** A cross-sectional online survey was conducted among health professionals who treat people in hospital following fractures. Respondents were asked about the use of care training for friends/family members of people discharged from hospital after fracture, and whether a clinical trial would be useful to test such carer training interventions. **Results:** A total of 114 health professionals accessed the survey. Fifty respondents (44%) reported that carer training was not offered in their practice. When it was offered, respondents reported this was not consistently provided. Less than 12% of respondents reported offering carer training to most of their patients following a fracture. What was offered in these instances was largely based on education provision (69%), practical skills in exercise prescription (55%) and manual handling (51%). Ninety-eight percent of respondents reported that a clinical trial would be, or would potentially be, valuable to aid a change in practice to include carer training in routine clinical care. **Conclusions:** Carer training programmes are not routinely provided in clinical practice for people following a fracture. The results indicate that health professionals see a potential value in these programmes, but further research is recommended to provide an evidence base for these interventions.

## 1. Introduction

Falls are a major problem for older people. Around a third of people aged 65 and over, and around half of people aged 80 and over, fall at least once a year [1]. Over three million people in the UK have osteoporosis. These individuals are at much greater risk of fracture after a fall [1]. The total annual cost of fractures in older people for the UK is estimated at GBP 4.4 billion, including GBP 1.1 billion for social care [1].

Older people who sustain a fracture can have a long and difficult recovery after hospital discharge [2]. Hip, pelvis, wrist, humerus and spinal fractures, or a combination of these, i.e., hip and wrist, can have long-term consequences. For example, after hip fracture, approximately 77% of people never return to pre-injury levels of function and independence [3]. Quality of life is reduced, and mortality is high [4]. People affected often experience continued falls and re-injury. This leads to reduced independence and diminishing confidence in self-caring and living at home.

During a hospital stay, older people who have sustained a fracture receive rehabilitation to improve their safety and independence in walking and self-care [5]. On discharge from an inpatient setting, people can receive no rehabilitation or very limited rehabilitation of variable quality [6,7]. Family members and friends, acting in the role of unpaid carers, commonly support patient recovery [8]. Tasks that they may assist with range from personal activities of daily living (ADL) such as toileting, washing, dressing and cooking, to more complex tasks such as managing money, shopping and household chores [9]. Caring is heterogeneous in terms of who undertakes the care, e.g., spouse, children, wider family and friends, and in what roles they adopt [10].

Caring and supporting an individual during and after a hospital admission for a fracture can be difficult. These individuals frequently experience physical and mental stress when trying to support their friend’s/family member’s recovery [2]. One in four carers for older people are aged over 65 years themselves [11]. These people have their own health challenges which can make caring roles even more difficult. Qualitative evidence suggests that although these people want to support their friends/family members, they are frequently under-skilled and have low confidence in doing so [12]. Teaching friends and family members who will be carers the skills to better support older people discharged from hospital following a fracture may improve physical function, independence and quality of life and reduce the requirement for other health and social services. By better equipping carers and patients with knowledge about their recovery and rehabilitation, older people’s recovery can be optimised, rather than guided with the ‘trial and error’ approach which currently exists. For carers, it may reduce the anxieties associated with their role and increase their quality of life.

### Purpose

There is uncertainty within the literature on what health professionals in hospitals do to support family members or friends who act as unpaid carers during the hospital to home transition for older people who experience a fracture. It is not clear if they offer support, education or training. If they do, it is not clear who provides this, when, where, or how. It is also unclear whether clinical trial evidence would help health professionals to implement a carer training programme for older people following a fracture. Previous literature has indicated a need for patient-related home care knowledge to carers prior to hospital discharge [13,14], but it is unclear if this happening. Given this uncertainty and the lack of evidence reporting on current practice in the hospital to home transition, the purpose of this survey was to provide novel insights to determine the current practice in these respects.

## 2. Materials and Methods

This study has been reported in accordance with STROBE [15] and CHERRIES [16].

### 2.1. Design

This is an online, cross-sectional, survey.

### 2.2. Participants and Approach

Two approaches were adopted to identify potential respondents using a convenience sampling approach. Firstly, we used a social media platform (X/Twitter) to promote the survey. Tweets were sent from the investigators’ X accounts (@tobyosmith; @FinneganSusanne). The tweets included the survey hyperlink where potential respondents accessed the Participant Information Page, Consent Page and Survey (Appendix A). The second approach was through professional networks. The Association of Trauma and Orthopaedic Chartered Physiotherapists (ATOCP), AGILE (the Professional Network for Physiotherapists working with Older People), the British Geriatric Society (BGS) and the International Association of Physiotherapists Working with Older People (IPTOP) organisations acted as survey gatekeepers. They distributed the survey to their membership through an email/text.

### 2.3. Potentially Eligible Participants

Practising health professionals including (but not exclusive to) the following: members of the physiotherapy, occupational therapy or nursing teams, medical practitioners and surgeons.Those working in trauma and orthopaedic, musculoskeletal, geriatric/older people’s/frailty or rehabilitation services.The aforementioned participants provided informed consent in order to participate in the study.

We excluded individuals who were unable to complete the online survey web form.

Those potential participants approached to complete the survey through social media or gatekeeper channels were directed to a Qualtrics survey webpage. They were provided with further information about the survey study in the form of a Participant Information Page. Consent was recorded by ticking a box at the end of the Participant Information Page, prior to the first survey questions.

### 2.4. Data Collection

The survey comprised 13 questions. These explored the following aspects: whether carer training programmes are delivered by health professionals and, if so, who these are for, who delivers them, when, where and how. If carer training programmes were delivered, we asked if the respondents felt they were beneficial, for whom, when and how. Finally, we explored whether health professionals felt a carer training programme could be delivered in their service and whether clinical trial evidence would provide value in brokering practice change. The survey provided partial closed-ended questions requiring a categorical response. The survey was piloted with three health professionals (physiotherapist, nurse, occupational therapist) and two patient–carer dyads who had experience in the recovery of a hip fracture. All were UK-based. This ensured that the survey was acceptable and clear. It was also ensured that the survey would take a maximum of 10 min to complete, which was explained in the main survey’s opening Participant Information Sheet.

To promote a high response rate, the social media channels used for recruitment were harnessed to encourage completion, with weekly posts and messaging to make sure that the surveys were delivered during the six-week data collection window.

### 2.5. Sample Size

There is no consensus on the optimal sample size for an online survey of this nature. Due to an absence of literature on the number of older people who sustain fractures and have a nominated carer within specific countries, it was not possible to determine the potential response rate and cohort size for this survey. The sample size was based on the study time frames. We opened the survey for a six-week period (2 September 2024 to 18 October 2024) to gain the largest and most representative sample that was feasible.

### 2.6. Data Analysis

The primary analysis was to assess responses related to the use (or non-use) of carer training programmes by people supporting older people when discharged from hospital after a fracture. Accordingly, frequency distributions and percentages were used to collectively assess all the completed surveys.

We planned secondary analyses, including categorising responses by the following classifications: (1) professional role; and (2) geographical location. However, as the largest proportion of responses were derived from physiotherapists and UK clinicians, the representation from other groups was insufficient to offer meaningful comparisons (Table 1; Figure 1). These were, therefore, not performed.

## 3. Results

### 3.1. Respondent Characteristics

In total, 114 respondents were eligible and provided data for the survey. The majority of respondents were physiotherapists (*n* = 78; 68%), with three occupational therapists (3%), two orthopaedic surgeons (2%) and a single nurse, physician, orthopaedic theatre support worker and pharmacy representative (1% each). Twenty-seven (23%) respondents did not state their professional role.

Eighty-seven respondents provided data on their country of origin. Figure 1 illustrates the global representation of respondents, with the majority being from the UK (*n* = 57; 65%). In total, 75 respondents (66%) reported that they provided carer training, guidance or education to family members or friends who were expected to support older people at home following a fracture.

### 3.2. Delivery of Carer Training for People Following Fracture

When asked if carer training is currently offered to people following a fracture, 50 respondents (44%) reported that this was not offered. When offered, this was not consistent. Between 8 and 12% of respondents reported offering carer training to the majority of their patients following fracture (Figure 2).

Table 1 illustrates the responses related to what is currently provided to friends/family members in preparing them for caring for someone after a fracture. The majority of health professionals involved in carer training delivery were physiotherapists (always: 77%), occupational therapists (always: 31%), the general multidisciplinary team (always: 21%) or nursing team members (always: 14%). Professional groups who were less frequently involved in this included social workers (never: 58%), surgeons (never: 52%) or physicians (never: 37%).

Respondents were unable to identify a patient group where carer training was most required with the proposed universal coverage (Table 1). However, there was a suggestion that carer training should be delivered either in the hospital (always: 38%) or out of the hospital (always: 46%) rather than a combination of both (always: 24%). This was followed by a preference for face-to-face training in most opportunities (always: 71%) rather than virtual (always: 3%) or hybrid (always: 0%).

When asked about the constituents of a carer training programme, this most frequently included the provision of advice and guidance on recovery expectations (always: 69%), signposting to other services (always: 28%) and the provision of contact details for further support (always: 43%), in addition to education on potential complications (always: 43%). Twenty-four respondents (51%) reported always practising manual handling skills such as transfers and gait practice when they delivered carer training, with 55% reporting always providing exercise prescription and progression advice. Similarly, guidance on pacing and behaviour modification was always provided by 20 respondents (43%). Less frequently provided was advice on stress and anxiety management for carers (always: 19%).

When asked about the timing and duration of carer training, earlier provision was most frequently reported, with 46% reporting always providing their carer training in the inpatient setting or up to two weeks post-hospital discharge (31%). However, a proportion of respondents reported providing their carer training for up to six weeks post-hospital discharge (always: 24%) or longer (18%).

The respondents who reported that they do not provide carer training in their practice were asked who they would provide this to if they did offer it. As Table 2 illustrates, there was no specific target group of potential patients identified, with the respondents reporting consistent support for patients, irrespective of fracture type. However, this was based on only six responses from the survey cohort.

### 3.3. Factors Influencing Implementation

The respondents were asked if they could currently implement a carer training intervention into their clinical practice. Of these, 46 respondents (96%) reported that they could adopt or could potentially adopt this today, whereas 2 respondents (4%) reported that they would be unable to do so.

All but one respondent reported that evidence from a clinical trial would be or would potentially be valuable to help implement a carer training programme (yes: *n* = 27; potentially: *n* = 14) whilst one respondent (2%) reported that a trial would not be valuable. Similarly, all but one respondent (2%) reported that understanding the effectiveness of a carer training programme for older people after a fracture would be (67%) or would potentially be (31%) a critical area of research.

## 4. Discussion

The aim of this study was to determine what health professionals in hospitals do during the transition from hospital to home to support family members or friends who act as unpaid carers for older people who experience a fracture, and their views on what factors may influence carer training for older people following a fragility fracture. The findings from this survey indicate that carer training for older people in hospital with a fracture in preparation for discharge is not consistently provided by health professionals. When it is offered, this is the exception rather than the norm. There is agreement that this could be beneficial for older people irrespective of fracture type, with there being an agreement of its value for older people who live in the community, rather than in care homes. Key professionals involved in this include allied health professionals and nursing staff, but the respondents indicated that carer training could be delivered by any member of the multidisciplinary team. Key elements of a carer training programme suggested by respondents were education, advice and guidance on recovery expectations, ADLs, mobility and exercise and pacing and behaviour modification. There is a clear suggestion that research on carer training for older people following a fracture is warranted, as this could inform a change in practice if shown to be effective.

The survey indicates that training carers for older people during the transition following hospital discharge after a fracture is not routine practice. Whilst there is evidence that older people and their carers can be anxious about this transition from hospital discharge to early recovery [17,18], the provision of education, guidance and skills training to overcome this is not offered. Previous literature in other populations has highlighted that the provision of education on basic recovery skills around supporting the independence of ADLs whilst also highlighting potential complications or challenges in the first few weeks following hospital discharge can be helpful following stroke [19,20], cardiac surgery [21] and after other medical events [22,23]. This evidence remains limited in the fracture population, with only one hip fracture study currently published [24]. The survey was unable to identify why carer training is not routinely provided in hospitals. Previous studies have suggested that barriers could include resource limitations, financial constraints or a lack of awareness among healthcare professionals [25,26]. A formal evaluation may be instigated to better understand this, particularly at a service- and country-specific level. Nonetheless, given the poor functional outcomes which some older people experience following a fracture [27] and the anxieties which both patients and carers can exhibit during hospital discharge [18], a better understanding of the strategies needed to address this, such as carer training intervention, is warranted.

The respondents reported positivity towards the use of clinical trials to evaluate carer training interventions for older people following a fracture during the hospital discharge period. This reflects other studies reporting the value of trial evidence to foster clinical practice change [28]. The survey indicates that such a trial should be focused on older people following any fracture, as there was no clear indication that one group of patients, such as the hip fracture group, would differentially gain or not in a carer training intervention. These responses suggest that there are shared challenges in the discharge and early recovery of older people following a fracture. Based on the components suggested within carer training offered, for example, including information on recovery expectation, functional support with ADLs, guidance on pacing and behaviour modification and exercise prescription, the ability to test an intervention which could be flexible for a number of different patient groups is advantageous in both its potential value to the wider orthopaedic patient population, but also its in implementation to scale. Accordingly, studies such as HIP HELPER [24] should be reflected upon and a refocusing towards a wider population may be warranted.

This study has both strengths and limitations. The first strength includes the pre-survey submission pilot work with patients and carers in addition to clinicians. This provided support for the face validity of the survey and offered some confidence that the potential responses were plausible and relevant before the main respondents had an opportunity to complete the survey. Secondly, the open responses offered throughout the survey meant that unexpected responses could have been obtained. However, some limitations were evident. Firstly, the survey, whilst designed to capture international perspectives, was largely UK-based, and was relatively small for an international survey, potentially because of the sampling approach. Therefore, it was not possible to perform subgroup analyses of international responses. Similarly, some survey responses, such as determining the types of older patients who should be offered carer training interventions, were completed by a small number of participants (*n* = 6), with a high number of missing data for that component. Further consideration on how these results reflect on practices globally, with a larger number of diverse perspectives, would be valuable. Secondly, the largest professional group were physiotherapists, with minimal responses gained from other important professional groups such as occupational therapists, nurses and social workers. Gaining these perspectives and reflecting on how they may differ from physiotherapists’ would be valuable to understand both the perceived care needs that multiple professionals think older patients and carers need following a fracture and how professional identity and job role may impact on these perspectives. Thirdly, due to an absence in the literature on the number of older people who sustain fractures and have a nominated carer within specific countries, it was not possible to determine the potential response rate and cohort size for this survey. It is therefore not possible to determine if selection bias in respondents occurred and whether this impacted the generalizability of the findings. Finally, this survey was designed to understand clinical practice rather than patient and carer needs. There has been limited evidence from the fracture population on the perspectives of carers and older patients regarding the early recovery period. Better understanding this and, in particular, considering what could be offered to support this in the future from a patient perspective through qualitative research would be valuable and could influence future research and clinical application, with shared decision-making between health professionals and patients–carers regarding carer support for older people who have been discharged from hospital following a fracture.

To conclude, carer training during hospital discharge for friends/family members who support older people following a fracture does not consistently happen. Further evidence is recommended to firstly determine when carer training should be provided, who should deliver this, what should be included in a carer training intervention and how effective it is in improving both patient and carer outcomes for older people following a fracture, particularly in relation to timing, dosage and adherence to contact between patients–carers during early recovery. Given the poor clinical outcomes which some patients can experience following a fracture, and the anxiety and lack of support that carers who support these people once they are discharged receive, this may now be considered a research priority in older people’s orthopaedic rehabilitation and recovery.

## Figures and Tables

**Figure 1 geriatrics-10-00036-f001:**
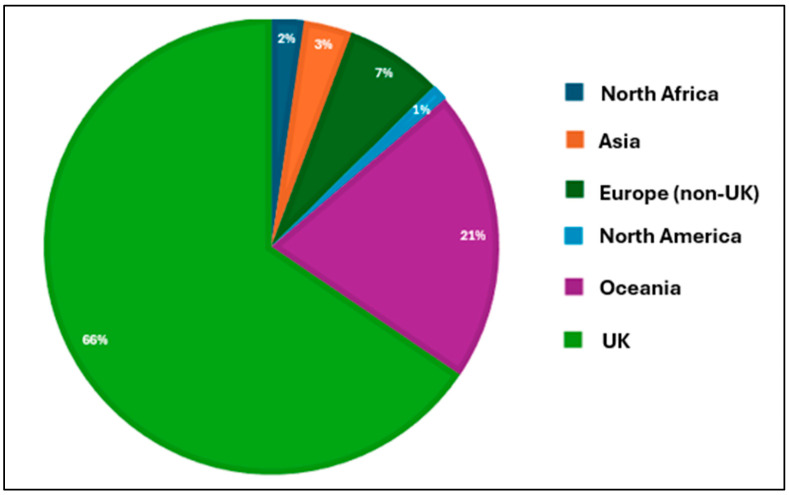
Country of origin for respondents (*n* = 87).

**Figure 2 geriatrics-10-00036-f002:**
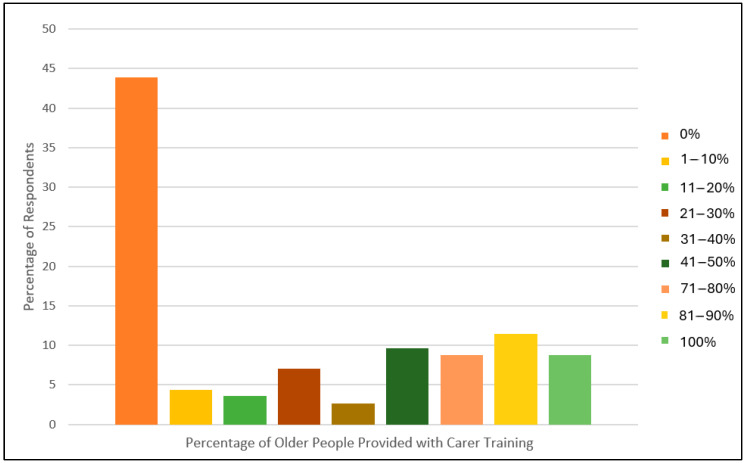
Bar chart illustrating the percentage of older people that the respondents reported as treating at home following a bone fracture.

**Table 1 geriatrics-10-00036-t001:** Summary of respondent’s answers to current carer training provision parameters.

	Always	Sometimes	Never	*N*
Which health professionals provided training for the respondents?				
Nursing Team	7 (14.3)	29 (59.2)	13 (26.5)	49
Physiotherapy Team	44 (77.2)	13 (22.8)	0 (0.0)	57
Occupational Therapy Team	16 (30.8)	31 (59.6)	5 (9.6)	52
General Multidisciplinary Team	10 (21.3)	30 (63.8)	7 (14.9)	47
Social Workers	1 (2.3)	17 (39.5)	25 (58.1)	43
Physicians	3 (7.0)	24 (55.8)	16 (37.2)	43
Surgeons	2 (4.7)	19 (44.2)	22 (51.2)	43
Other: Nursing Assistant, Pharmacist, Volunteer	1 (3.8)	4 (15.4)	21 (80.8)	26
What type of patients did the carers receive training for?				
Hip fracture	27 (54.0)	21 (42.0)	2 (4.0)	50
Other lower limb fracture (non-hip)	18 (36.0)	31 (62.0)	1 (2.0)	50
Spinal fracture	21 (42.0)	25 (50.0)	4 (8.0)	50
Upper limb fracture	16 (32.0)	32 (64.0)	2 (4.0)	50
Major trauma fracture	22 (45.8)	21 (43.8)	5 (10.4)	48
Patients who live at home with identified carer	27 (56.3)	19 (39.6)	2 (4.2)	48
Other: Caregivers, Care home carers, Rib fractures	2 (10.0)	8 (40.0)	10 (50.0)	20
Where is training provided?				
In hospital	15 (37.5)	18 (45.0)	7 (17.5)	40
Out of hospital	17 (45.9)	17 (45.9)	3 (8.1)	37
In and out of hospital	9 (24.3)	21 (56.8)	7 (18.9)	37
How is training provided?				
Face-to-face	34 (70.8)	13 (27.1)	1 (2.1)	48
Virtual (telephone or online)	1 (2.5)	22 (55.0)	17 (42.5)	40
Hybrid (face-to-face AND virtual)	0 (0.0)	15 (40.5)	22 (59.5)	37
What is provided in the carer training?				
Discussion on recovery expectations	31 (68.9)	13 (28.9)	1 (28.9)	45
Practice manual handling (transfers/walking practice)	24 (51.1)	22 (46.8)	1 (46.8)	47
Exercise prescription and progression advice	26 (55.3)	20 (42.6)	1 (42.6)	47
Signposting to other post-discharge services	13 (27.7)	32 (68.1)	2 (68.1)	47
Education on potential post-discharge complications	20 (42.6)	25 (53.2)	2 (53.2)	47
Pacing and behaviour modification advice	20 (42.6)	23 (48.9)	4 (48.9)	47
Advice on stress and anxiety management for caregivers	9 (19.1)	27 (57.4)	11 (57.4)	47
Provision of contact details for further support	20 (42.6)	24 (51.1)	3 (51.1)	47
Provision of written information/materials or online links	19 (40.4)	25 (53.2)	3 (53.2)	47
Duration of carer training provided				
Inpatient only	16 (45.7)	15 (42.9)	4 (11.4)	35
Up to 2 weeks post-hospital discharge	12 (30.8)	18 (46.2)	9 (23.1)	39
Up to 4 weeks post-hospital discharge	9 (23.1)	20 (51.3)	10 (25.6)	39
Up to 6 weeks post-hospital discharge	10 (24.4)	19 (46.3)	12 (29.3)	41
Longer than 6 weeks post-hospital discharge	8 (18.2)	21 (47.7)	15 (34.1)	44

**Table 2 geriatrics-10-00036-t002:** The frequency of respondents who recommended the types of older patients who should be offered carer training interventions to if were able to provide them.

	Strongly Agree	Agree	Neutral	Disagree	Strongly Disagree	*N*
Patients living alone	2 (33.3)	1 (16.7)	3 (50.0)	0 (0.0)	0 (0.0)	6
Patients living with an unpaid carer	4 (66.7)	1 (16.7)	1 (16.7)	0 (0.0)	0 (0.0)	6
Patients living in a care home/care facility	2 (33.3)	1 (16.7)	2 (33.3)	1 (16.7)	0 (0.0)	6
Patients following hip fracture	3 (50.0)	2 (33.3)	1 (16.7)	0 (0.0)	0 (0.0)	6
Patients following lower limb fracture (non-hip)	3 (50.0)	1 (16.7)	2 (33.3)	0 (0.0)	0 (0.0)	6
Patients following spinal fracture	2 (33.3)	3 (50.0)	1 (16.7)	0 (0.0)	0 (0.0)	6
Patients following multiple fractures	3 (60.0)	1 (20.0)	1 (20.0)	0 (0.0)	0 (0.0)	5
Major trauma patients	4 (66.7)	1 (16.7)	1 (16.7)	0 (0.0)	0 (0.0)	6
Any patient who has experienced a fracture	3 (50.0)	1 (16.7)	2 (33.3)	0 (0.0)	0 (0.0)	6

## Data Availability

Anonymised data will be made available on reasonable request to the corresponding author.

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
