# Peer review of "Health Professional Support for Friends and Family Members of Older People Discharged from Hospital After a Fracture: A Survey Study"

_geriatrics, 2025, doi:10.3390/geriatrics10020036_

Round 1
Reviewer 1 Report
Comments and Suggestions for Authors
This study reports the survey responses of 114 health professionals on provision of carer training for patients recovering from fracture. Despite the rationale for its use the survey confirmed that carer training is not part of usual care after fracture. As such the survey has identified a gap for future research.
Some specific comments:
Introduction line 57: Suggest amend to ‘…during and after hospital admission.’
Design line 80: Could the study be reported consistent with a suitable checklist e.g. CHERRIES
Table 2 and accompanying text is confusing. In the text it is stated that there were only two surgeon respondents but in the first section of Table 2 there are 43 responses assigned to surgeons. On viewing the questions I think Table 2 relates to the question about which members of your team provide training, not whether they provide it themselves, but it needs to be clarified.
Limitations, about line 245: It was not possible to report a response rate (as recommended in checklists). Therefore, it is possible there was a self-selection bias in that those who responded may have had more of an interest in the topic of carer training. Thus reinforcing the findings that carer training in practice may in fact be less than the low levels reported in this survey.
Limitations, line 257: It is acknowledged that the perspectives of carers and older patients were not sought. However, it is surprising that health professionals did not think it was an important part of the process to involve the patients I and their carers in decision making – whether or not they want to be involved in carer training is important factor for health professionals to consider.
Author Response
Comment 1: Fill out the STROBE checklist and attach it as supplementary to the article. https://www.equator-network.org/reporting-guidelines/strobe/
Response: We have included this as a supplementary file but have not cross-referenced this into the paper. If this is desired, please say and we will happily do this.
Comment 2: Information about the research population is missing. Who have been them? How many have been them.
Response: We have provided information on the target population in Paragraph 2 of the Introduction (Introduction, Lines 48-56). The eligibility criteria (Methods, Section 2.3 Line 92-99). There is insufficient data from registries or the literature to determine what the potential cohort size is and therefore we were unable to determine the response rate. This has now been explicitly stated in the paper (Methods, Section 2.5, Line 124-126). WE hope this addresses the comment raised.
Comment 3: Did you validate the data collection tool? Pilot-tested? Checked reliability?
Response: The survey was pilot-tested with three health professionals (physiotherapist, nurse, occupational therapist) and two patient-carer dyads who had experience of the recovery of a hip fracture. This has been presented in the Methods section (Methods, Section 2.4, Lines 112-115). We did not do any reliability checking statistically given the number of pilot participants and the purpose of this was to assess validity only. The results of the pilot are more clearly presented in this revision (Methods, Section 2.4, Lines 115-118).
Comment 4: What strategies did you use to improve the response rate? Reminders?
Response: Social media messaging was used to promote response rate and survey engagement. This is now stated more clearly in the revised paper (Methods, Section 2.4, Lines 119-121).
Comment 5: Start the discussion section with the research aim.
Response: We have now included this in the Discussion (Discussion, Paragraph 1, Line 201-204).
Reviewer 2 Report
Comments and Suggestions for Authors
Introduction
It would be beneficial to include references to previous studies that have evaluated post-discharge home care following a fracture. Citing these studies would clarify what is already known and what gaps remain in the literature. Additionally, the authors should clearly define the novelty of this study.
Results
The beginning of the Results section presents the breakdown of respondents’ professions. However, while 117 respondents participated, only 87 are accounted for in the profession breakdown. Are the remaining 27 classified under “Other”? If so, this should be explicitly stated in the text.
Discussion
The argument that carer training should be provided is reasonable, and the discussion on the need for clinical trials is clear. However, the study lacks a detailed analysis of why carer training is not sufficiently provided. Identifying barriers such as resource limitations, financial constraints, or a lack of awareness among healthcare professionals would strengthen the discussion.
Overall Assessment
This study provides an interesting examination of the provision of carer training for older adults following a fracture. However, its primary limitation is the lack of practical implications. Specifically, it does not adequately address when carer training should be provided, who should deliver it, what specific training should be included, or how training impacts patient outcomes.
Additionally, the study does not assess whether carer training improves patient outcomes, making it difficult to establish the value of such training. Without evidence of its impact, the practical significance of this study is unclear.
Furthermore, the total sample size is small, making it difficult to draw statistically significant conclusions. While the study is described as an “international survey,” 65% of the respondents are from the UK, limiting its global applicability.
Author Response
Comment: Introduction: It would be beneficial to include references to previous studies that have evaluated post-discharge home care following a fracture. Citing these studies would clarify what is already known and what gaps remain in the literature. Additionally, the authors should clearly define the novelty of this study.
Response: Thank you. We have cited previous literature around post-discharge home care (Reference 5, 13, and 14) and offered greater detail on the novelty of the study (Introduction, Purpose, Lines 76-81)
Comment: Results: The beginning of the Results section presents the breakdown of respondents’ professions. However, while 117 respondents participated, only 87 are accounted for in the profession breakdown. Are the remaining 27 classified under “Other”? If so, this should be explicitly stated in the text.
Response: Apologies that this was not sufficiently clear in the original submission. Twenty-seven respondents did not state this. We have therefore acknowledged this in the paper for clarity (Results, Section 3.1, Lines 158-159).
Comment: Discussion: The argument that carer training should be provided is reasonable, and the discussion on the need for clinical trials is clear. However, the study lacks a detailed analysis of why carer training is not sufficiently provided. Identifying barriers such as resource limitations, financial constraints, or a lack of awareness among healthcare professionals would strengthen the discussion.
Response: Thank you for highlighting this very important point. We have provided greater clarity on this within the revised Discussion (Discussion, Paragraph 2, Lines 243-248).
Comment: Overall Assessment: This study provides an interesting examination of the provision of carer training for older adults following a fracture. However, its primary limitation is the lack of practical implications. Specifically, it does not adequately address when carer training should be provided, who should deliver it, what specific training should be included, or how training impacts patient outcomes.
Response: This is important, and we have highlighted the need to better-answer some of these important uncertainties in the concluding paragraph (Discussion, Paragraph 5, Lines 295-302).
Comment: Overall Assessment: Additionally, the study does not assess whether carer training improves patient outcomes, making it difficult to establish the value of such training. Without evidence of its impact, the practical significance of this study is unclear.
Response: This point is related to that directly above. We agree with the value of emphasising this for the reader and have therefore revised the conclusion to stress this point (Discussion, Paragraph 5, Lines 295-302).
Comment: Overall Assessment: Furthermore, the total sample size is small, making it difficult to draw statistically significant conclusions. While the study is described as an “international survey,” 65% of the respondents are from the UK, limiting its global applicability.
Response: We agree and have highlighted this limitation further in the revised Limitations section (Discussion, Paragraph 4, Line 273).
Reviewer 3 Report
Comments and Suggestions for Authors
The authors present an interesting article titled: “Health Professionals Support to Friends and Family Members of Older People Discharged from Hospital After Fracture: An International Survey”.
The authors address a common and highly relevant topic in daily clinical practice: fractures in older individuals, which can be associated with functional decline, increased mortality, and reduced quality of life. For example, as mentioned in the introduction (lines 43-44) of the manuscript, a significant proportion of older patients who suffer a hip fracture do not regain their pre-injury functional status. Training family members through health professionals to support their relatives who have suffered a fracture could be an approach for optimizing care after hospital discharge.
Comments and suggestions:
- The introduction is concise and effectively presents the key facts related to the research project.
- The detailed information on potential study participants in section 2.3 is an important aspect, as it provides a clear overview of the study’s target group.
- In Figure 1, adding percentage values to the diagram would be beneficial. In its current form, the pie chart only allows for a rough estimate.
- A key issue with the study is the superficial nature of the questions Q2.7 and Q2.8. More detailed information on the specific training measures and their extent would be desirable, as significant variations may exist. While the study notes that the training was conducted during hospitalization, for 2, 4, 6, or more than 6 weeks, the intensity of the training remains unclear. For example, daily sessions over 2 weeks could potentially result in a greater training effect than weekly sessions over 4 weeks. However, this aspect also highlights the need for future studies to evaluate the effectiveness of standardized training programs for family members.
- I appreciate the authors’ statement on the study’s limitations and their suggestions for future research.
Author Response
Comment: The authors present an interesting article titled: “Health Professionals Support to Friends and Family Members of Older People Discharged from Hospital After Fracture: An International Survey”.
Response: Thank you for your comments. We have addressed the points raised and itemised these below.
Comment: The authors address a common and highly relevant topic in daily clinical practice: fractures in older individuals, which can be associated with functional decline, increased mortality, and reduced quality of life. For example, as mentioned in the introduction (lines 43-44) of the manuscript, a significant proportion of older patients who suffer a hip fracture do not regain their pre-injury functional status. Training family members through health professionals to support their relatives who have suffered a fracture could be an approach for optimizing care after hospital discharge. Comments and suggestions:
Response: Thank you for your comments. We have addressed the points raised and itemised these below.
Comment: The introduction is concise and effectively presents the key facts related to the research project.
Response: Thank you. No amendment required.
Comment: The detailed information on potential study participants in section 2.3 is an important aspect, as it provides a clear overview of the study’s target group.
Response: Thank you. No amendment required.
Comment: In Figure 1, adding percentage values to the diagram would be beneficial. In its current form, the pie chart only allows for a rough estimate.
Response: We have now provided percentages to Figure 1 as recommended (Figure 1).
Comment: A key issue with the study is the superficial nature of the questions Q2.7 and Q2.8. More detailed information on the specific training measures and their extent would be desirable, as significant variations may exist. While the study notes that the training was conducted during hospitalization, for 2, 4, 6, or more than 6 weeks, the intensity of the training remains unclear. For example, daily sessions over 2 weeks could potentially result in a greater training effect than weekly sessions over 4 weeks. However, this aspect also highlights the need for future studies to evaluate the effectiveness of standardized training programs for family members.
Response: We acknowledge this is a limitation on what we could ask in the survey. However, to ensure the reader understand this, we have provided a clearer research-direction where these parameters are recommended to be assess in future trials. This is stated in the revised Discussion section (Discussion, Paragraph 5, Lines 295-302).
Comment: I appreciate the authors’ statement on the study’s limitations and their suggestions for future research.
Response: Thank you. No amendment required.
Round 2
Reviewer 1 Report
Comments and Suggestions for Authors
The authors have addressed the minor concerns of the authors resulting in an improved manuscript that will make a contribution to the literature.
Author Response
Thank you for your kind comments. There is nothing further to address in relation to Reviewer 1's comments.
Reviewer 2 Report
Comments and Suggestions for Authors
The authors have made thoughtful revisions, and the manuscript has improved as a result. I appreciate their effort in addressing the feedback and clarifying key aspects of the study.
However, a few concerns remain. Calling this an “International Survey” may be misleading, as the sample size is limited and most respondents are from the UK. I suggest revising the title to better reflect the sample or adding a clarification in the abstract and discussion. Additionally, Table 2 (N=6) is based on a very small sample, so it should either be removed or clearly labeled as exploratory to prevent misinterpretation.
Author Response
Comment: However, a few concerns remain. Calling this an “International Survey” may be misleading, as the sample size is limited and most respondents are from the UK. I suggest revising the title to better reflect the sample or adding a clarification in the abstract and discussion.
Response: As recommended we have removed the term ‘International’ from the Title (Title, Page 1) and paper (Abstract, Methods, Line 15; Materials & Methods, Design, Line 15).
Comment: Additionally, Table 2 (N=6) is based on a very small sample, so it should either be removed or clearly labeled as exploratory to prevent misinterpretation.
Response: We have included these data as this was the intention to report. However, we have acknowledged this limitation in both the Results (Results, Lines 202-203) and Discussion’s Limitation (Discussion, Lines 275-279).
Reviewer 3 Report
Comments and Suggestions for Authors
The authors of the manuscript: "Health Professionals Support to Friends and Family Members of Older People Discharged from Hospital After Fracture: An International Survey" have revised and improved their paper. I have no further points of criticism.
Author Response
Thank you for your kind comments. There is nothing further to address in relation to Reviewer 3's comments.